

# A conceptual replication of the Psychological Typhoon Eye effect in the aftermath of the Petrinja earthquake in Croatia

Gaëtan Mertens and Marta Dürrigl

Department of Medical and Clinical Psychology, University of Tilburg, Tilburg, Netherlands

## ABSTRACT

**Background:** The Psychological Typhoon Eye (PTE) effect refers to the observation that those living in the epicenters of natural disasters or public emergencies exhibit lower levels of psychological distress than those living further away. The effect has been described in the aftermath of multiple public emergencies, including the 2008 Wenchuan earthquake. However, despite its potential importance for emergency relieve, this phenomenon has received little research attention and requires further replication. The goals of this study were to replicate the PTE effect using both the original items used in prior research and using the validated Depression, Anxiety, and Stress Scales (DASS).

**Methods:** A cross-sectional survey was conducted following an earthquake with a magnitude of 6.4 on the Richter scale that occurred in December 2020 in Petrinja, Croatia. The sample consisted of 316 participants living in Croatia at the time of the earthquake. Questionnaires were administered through an online survey, including the DASS, the items used in previous research, and questionnaires measuring general and earthquake-specific coping. Exposure to the earthquake was operationalized as the degree of devastation participants had experienced (*i.e.*, the degree of structural damage to their home as assessed by local authorities). In line with previous work, we tested for an inverse relationship between the experienced level of devastation and the different measures of psychological distress (*i.e.*, the items used in previous work and the DASS).

**Results:** We found no evidence for the PTE effect in our study. Instead, we observed a ripple effect, whereby those most affected by the earthquake showed the most psychological distress. We argue that the ripple effect, rather than the PTE effect, should be seen as the default psychological response pattern to natural disasters and emergencies.

Corresponding author
Gaëtan Mertens,
Mertensgaetan@gmail.com

[1] Portions of this text were previously published as part of a preprint (https://doi.org/10.31234/osf.io/m9qnc).

# INTRODUCTION[1]

The Psychological Typhoon Eye (PTE) effect is a recently described pattern of public psychological responses to natural disasters and social emergencies (*Zhang et al., 2020*). In particular, the PTE refers to the observation that people living in the epicenters of natural disasters or public emergencies have lower anxiety levels (*Xie et al., 2011*) and less health and safety concerns (*Li et al., 2009*) than those living further away from these epicenters. The name for this phenomenon was based on its similarity to the seemingly calm "eye" of typhoons and hurricanes. The PTE phenomenon gained traction in recent years as it is in stark contrast to the previously established pattern of psychological responses to natural disasters and public emergencies called the "ripple effect". The ripple effect entails that the closer a person is to the epicenter of a devastating event, the more likely he or she will experience psychological consequences (*Slovic, 1987*; *Xie et al., 2011*; *Wen et al., 2020*). However, the conditions under which either the PTE effect or the ripple effect are observed are still unknown and require further research attention (*Wei et al., 2017*).

The PTE was first described in a study by *Li et al. (2009)*, who conducted a study following a disastrous earthquake in the Wenchuan region in China in 2008. The earthquake was of 8.0 magnitude on a Richter scale, and resulted in over 60,000 deaths, more than 350,000 injuries, and more than 17,000 people listed as missing. This initial study by *Li et al. (2009)* aimed to assess people's post-earthquake health and safety concerns. The authors hypothesized that these concerns would gradually vary depending on the devastation levels, expanding out from the epicenter of this natural disaster (*i.e.*, a ripple effect). *Li et al. (2009)* categorized the devastation levels in three categories: slightly devastated, moderately devastated, and severely devastated. Contrary to their hypotheses, those living in the epicenter of the earthquake, with the emphasis on those whose residential devastation levels were severe, showed fewer concerns regarding their health and safety compared to those who experienced less residential devastation or even those in non-devastated areas. Having identified and named this new phenomenon, they conducted follow-up surveys four and 11 months after the previous study to investigate the robustness of their previous findings and find potential explanations for why this phenomenon occurred (*Li et al., 2010*). In addition to administering the previous questions about health and safety concerns, in the following two data collection waves the researchers also inquired about the relational distance to the damage experienced, whereby the participants could indicate if they themselves, their family or friends suffered any damage or were physically hurt (*Li et al., 2010*). The results again showed that the level of concern did not grow with the relational distance to the physical or economic damage (*Li et al., 2010*), but instead followed the same trend as in the original study (*i.e.*, lower levels of concerns near the epicenter). The researchers concluded that the PTE effect remained robust throughout the entire year after the Wenchuan earthquake and that this has implications for providing psychological services to affected areas (*Li et al., 2010*).

Other studies have provided further evidence for the PTE phenomenon. In particular, in the context of the 2002–2004 SARS epidemic, it was shown that the proximity to the epicenter of the epidemic outbreak was associated with less anxiety experienced about

contracting the disease (*Xie et al., 2011*). A similar pattern was observed in response to the COVID-19 outbreak in Wuhan, China, with more mental health problems being reported in provinces with fewer COVID-19 cases (*Zhang et al., 2020*). Finally, studies on lead-zinc mining risk (*Zheng et al., 2015*) and public response to terrorism (*Li et al., 2020*) also provide more evidence in favor of the PTE phenomenon. Each of these studies showed that participants in the epicenter of a public emergency or a natural disaster, despite being closer to a potential risk, showed less psychological distress than participants further away from it.

However, one limitation common to most of these different previous studies is that they used custom-made scales to assess psychological distress, rather than validated questionnaires. For example, *Li et al. (2009)* used five custom-made questions to assess post-earthquake concerns such as "How many medical doctors are needed for every 1,000 residents in the earthquake areas?" and "Suppose that there is a medication that can heal the psychological wounds of mass disaster without side effects such as nausea or anaphylaxis. What dose of the medication should be prescribed for an earthquake victim (up to 100 mg daily)?". These questions are non-standard items to measure psychological distress, raising doubts about their psychometric soundness (*e.g.*, their convergent validity with scales for measuring psychological distress). Additionally, the use of idiosyncratic items instead of standardized questionnaires complicates any direct comparisons with other studies. Finally, the focus on medical and psychological help in these questions by *Li et al. (2009)* makes them difficult to appropriately answer by non-expert respondents. Hence, replication of the PTE is required using validated and standardized scales.

Another issue with the previous work is that they focused on different kinds of public emergencies. Arguably, an earthquake entails different risks than a pandemic outbreak or terrorist attacks. For example, pandemics often involve risks for specific demographic groups such as elderly or children (*Esai Selvan, 2020*), whereas an earthquake can in principle affect anyone. Furthermore, individuals affected by earthquakes most often live in geological unstable areas and hence are at a continuous risk to be exposed to an earthquake again. In contrast, pandemics can in principle start anywhere in the world and often follow a natural course, after which the risks are largely contained (*World Health Organization, 2009*). These differences between these public emergencies may therefore result in different psychological responses. So far, only two studies by the same author team have focused on the PTE effect after an earthquake. Therefore, the PTE effect in the aftermath of an earthquake requires further replication.

Theoretically, there are several competing explanations for the PTE effect (*Xu et al., 2020*). One proposed theory in previous studies is that the effect of exposure to a disaster or an emergency on mental health might be mediated by coping self-efficacy (*Zhang et al., 2020*). Coping self-efficacy refers to the perceived availability and effectiveness of behaviors necessary to avoid or mitigate a threat, also known as the response efficacy (*Maddux & Rogers, 1983*). According to this explanation, those most directly exposed to an emergency or natural disaster may feel that they acquired the necessary skills to mitigate these threats in the future, thereby improving their coping self-efficacy. Indeed, in a study on the PTE related to the COVID-19 pandemic outbreak, *Zhang et al. (2020)* found that coping

efficacy mediated the negative association between the level of exposure and mental health problems. However, it is currently unclear whether general coping self-efficacy or disaster specific self-efficacy (*e.g.*, relating to earthquakes, COVID-19, *etc.*,) mediates this relationship. This is an important question for providing focused psychological support in the aftermath of natural disasters. Hence, this mediating role for coping efficacy to explain the PTE effect requires additional research.

Taken together, the PTE effect has been observed in several studies following natural disasters and emergency situations but deviates from established patterns and common knowledge. However, most previous studies on the PTE effect have used custom-made scales for measuring psychological distress, highlighting the need to replicate this phenomenon using validated questionnaires. Furthermore, there is remaining uncertainty about the underlying mechanisms behind this pattern of stressful responding after natural disasters. Given the potential relevance of the PTE effect in providing psychosocial support following catastrophes (*e.g.*, determining which affected areas to prioritize), replicating this effect and examining its working mechanisms is important. An independent replication, particularly across different countries and cultural contexts, can demonstrate the robustness of this phenomenon. Furthermore, a better understanding of its working mechanisms is important for intervention studies (*e.g.*, psychosocial support with an emphasis on gaining self-efficacy). Finally, research with validated questionnaires is needed to establish whether the PTE effect can be observed using conventional operationalizations of psychological distress.

To address these literature gaps the aims of the current study were to replicate the Psychological Typhoon Eye effect in a new sample using validated scales and to investigate some of its potential working mechanisms. Therefore, we conducted a study in Croatia in the aftermath of the Petrinja earthquakes on the 29th of December 2020. With a magnitude of 6.4 on the Richter scale, the Petrinja earthquakes are among the most devastating natural disasters to occur Croatia in recent decades. Most affected regions included Sisak-Moslavina county, several other Croatian counties, and areas of neighboring Bosnia and Herzegovina, and Slovenia. According to official reports, seven people died and 26 were injured with six suffering serious injuries. The estimated material damage amounted to around 5.5 billion euros and the 2nd of January 2021 was pronounced as the National Day of Mourning in Croatia in honor of the victims of the earthquake. Our study was conducted within a year after the Petrinja earthquakes (*i.e.*, in December 2021). The aim of our study was to try to replicate the PTE effect both with the original questions developed by *Li et al. (2009)* and using the Depression, Anxiety, and Stress Scales (DASS; *Lovibond & Lovibond, 1995*). We hypothesized to replicate the PTE using the original questions used by *Li et al. (2009)*. Similarly, we hypothesized to replicate the PTE using the DASS. Finally, we expected that coping self-efficacy will mediate the relationship between the level of experienced devastation and psychological distress.

## MATERIALS AND METHODS

### Participants

We recruited a convenience sample of residents of Croatia. To be eligible for participation, participants had to be 18 years or older, present in Croatia at the time of the earthquake in December 2020 and be willing to take part in the study. We administered a written consent form. We used the software program G*Power (*Faul et al., 2007*) to conduct a power analysis to determine the required sample size. Our goal was to obtain 0.95 power to detect a Cohen's f medium effect size of 0.25 at the standard 0.05 alpha error probability. The sample size was calculated for the one-way analysis of variance (ANOVA), with level of devastation as an independent factor with four levels (four colored stickers: white, green, yellow, and red; see below). The required sample size according to this power analysis was 280 participants. Ethical approval was obtained from the Ethics Review Board of the faculty of the Tilburg School of Social and Behavioral Sciences (approval number: TSB_RP48).

To recruit Croatian respondents for this study, we collaborated with the Catholic University of Croatia. The collaboration entailed that several affiliated professors shared the questionnaire among university staff and students to recruit the required number of participants. Furthermore, participants were also recruited through social media by sharing the questionnaire through different groups on various platforms such as Facebook, Instagram, Reddit. Additionally, a radio interview was given on a Croatian radio station about the study and this station shared the questionnaire through the emailing lists of the interviewers and radio hosts.

During the initial sampling (November–December 2021), 315 participants who were in Croatia at the time of the Petrinja earthquake filled out the questionnaire. Additionally, eight more participants completed the questionnaire later (*i.e.*, in June 2022) to try and increase the number of respondents in the highest devastation category (see below). In total, 323 participants took part in this study. Seven of these participants were excluded from the study for improperly filling out the demographic data or being underaged ($<18$ years old). The final sample consisted of 316 participants, ($M_{age} = 36.77$ years, SD = 13.42) out of which 77.8% were female. The vast majority (97.8%) of our sample was Croatian, with three other participants indicating their nationalities were Bosnian, Slovenian, and Albanian. Seventy-seven (24.4%) of our participants obtained a bachelor's degree and 123 (38.9%) obtained a master's degree, whereas the remaining 36.7% of our sample had a lower education level.

We also assessed the level of devastation participants' residence suffered. Out of the 316 participants in the final sample, 151 (47.8%) reported having no damage to their homes, 134 (42.4%) reported having some light damage, 21 (6.6%) reported their homes were temporarily uninhabitable, and 10 (3.2%) reported their homes were severely structurally damaged and required immediate evacuation after the earthquake. Sixty-seven (21.2%) of our participants lived in the county where the epicenter of the earthquake occurred.

## Instruments

An online questionnaire using the Qualtrics software was distributed and consisted of two parts. The first part included questions regarding demographic data (*i.e.*, gender, age, education level, nationality, county of residence). The second part comprised of the Croatian version of the DASS-21 scale (*Ivaković, 2019*; *Lovibond & Lovibond, 1995*), five questions about post-earthquake concern from the original study by *Li et al. (2009)*, a Croatian version of the coping self-efficacy scale (*Ivanov & Penezić, 2002*; *Schwarzer, 1993*), earthquake-specific coping efficacy (*Sumer et al., 2005*), and lastly an earthquake risk perception measure (*Kung & Chen, 2012*).

### Level of devastation

The level of devastation to the residence of the participants at the time of the earthquake was assessed using colored stickers. In particular, immediately after the earthquake in Croatia, structural engineers assessed the risk and potential dangers of structurally impacted buildings. They have a system of colored stickers, added to the front part of the house, to indicate the severity of the damages. Stickers came in four colors: white, green, yellow, and red. White stickers were used if the buildings were not impacted at all. The green stickers indicate that there is no structurally significant damage to the building and there is no risk of the building collapsing. Yellow stickers indicate that the building is temporarily uninhabitable, meaning for the time being, living in these buildings can be a risk, however there is no need for an immediate evacuation. Finally, red stickers show the building is a hazard and poses a serious life risk. It is advised that the building is immediately evacuated because the structural damage is too severe to withstand another potential earthquake. This sticker system provides an ecologically valid way of operationalizing the level of devastation experienced by the respondents in our study. That is, the residents who received the red stickers can be taken as those participants who experienced the highest intensity impact of the Petrinja earthquake, while the other stickers indicate that the participants were gradually less affected by the earthquake.

### DASS-21

The Croatian version of the DASS-21 was used (*Lovibond & Lovibond, 1995*; *Ivaković, 2019*). The scale comprises 21 items, divided into three self-reported scales, designed to measure states of depression, anxiety, and stress. Responses are measured from zero to three, with zero meaning "Did not apply to me at all" and three meaning "Applied to me very much or most of the time". Participants must respond to how each statement applied to them over the past week. An example of an item measuring a state of depression is "I couldn't seem to experience any positive feeling at all". An item measuring state of anxiety is "I experienced trembling (*e.g.*, in the hands)" and an item measuring the state of stress is "I found myself getting agitated". The DASS-21 shows good reliability, with Cronbach's alpha values of 0.84, 0.83 and 0.87 for the three subscales measuring the states of depression, anxiety, and stress, respectively (*Ivaković, 2019*).

### Post-earthquake levels of health and safety concern

The post-earthquake concern was measured by five questions from the original study by *Li et al. (2009)*. These five questions aimed at assessing the post-earthquake concern about safety and health. The questions were the following: "What is the probability (0–100%) that an epidemic disease will be widespread in the post-earthquake areas?" (Illness probability); "How many times (out of 100 aftershocks) would residents in the earthquake areas need to take safety measures?" (Safety measures); "How many medical doctors are needed for every 1,000 residents in the earthquake areas?" (No. doctors); "How many psychological workers are needed for every 1,000 residents in the earthquake areas?" (No. psychologists) and "Suppose that there is a medication that can heal the psychological wounds of mass disaster without side effects such as nausea or anaphylaxis. What dose of the medication should be prescribed for an earthquake victim (up to 100 mg daily)?" (Medicine dose).

### Coping self-efficacy

An adapted and translated coping self-efficacy scale (*Schwarzer, 1993*; *Ivanov & Penezić, 2002*) was used. The scale comprises 10 items, measuring the perceived self-efficacy when one is faced with various challenges. Responses are measured from one to five, with one indicating "not true at all" and five indicating "exactly true". Participants are prompted to respond based on how they perceive themselves. Examples of items include "I am confident that I could deal efficiently with unexpected events", "I can solve most problems if I invest the necessary effort" and "I can always manage to solve difficult problems if I try hard enough". The overall score is created by adding up the item responses. The self-efficacy scale shows good reliability, with Cronbach's alpha value of 0.85 (*Ivanov & Penezić, 2002*).

### Earthquake-specific coping self-efficacy

Coping efficacy was examined in the context of earthquakes using the four-item earthquake-specific self-efficacy scale adapted by *Sumer et al. (2005)*. The items are "I believe that I will overcome the difficulties of this earthquake experience", "I have the resources and belief I need to successfully handle this earthquake experience", "I'm able to think about the earthquake and those I lost more comfortably" and "I believe that my daily life has normalized". Responses are measured on a three-point scale and *Sumer et al. (2005)* found an internal consistency of Cronbach's alpha = 0.74.

### Earthquake risk perception

Lastly, the measures included the earthquake risk perception measure (*Kung & Chen, 2012*). The scale comprises eight questions, with responses measured from one (not at all) to four (very much). Examples of items measuring earthquake risk perception are "How much do you think your life is threatened by earthquakes? "How likely do you think it is that earthquakes will hit the place you live? "and "How much are you aware of the preparedness measures you can take before the earthquakes?". The observed internal

consistency for this scale was Cronbach's alpha = 0.59. However, because we had not formulated any specific hypotheses relating to this scale, it was not considered further in the data analysis.

### Questionnaire translations

The post-earthquake levels of health and safety concern (*Li et al., 2009*), earthquake-specific coping self-efficacy (*Sumer et al., 2005*), and earthquake risk perception (*Kung & Chen, 2012*), questionnaires did not have previously validated translation in Croatian and were therefore translated to Croatian by the first author, who is a native speaker, and independently checked by two psychology master students at the Catholic University of Croatia. Each student translated the questionnaire independently and the final translation decision was made by comparing the translations and reaching a mutual decision.

## Procedure

The questionnaires were combined into an online survey using the Qualtrics platform and were distributed through the snow-balling technique by sharing it on social media profiles, Facebook student groups, Facebook groups specifically made to help in the wake of the 2020 earthquake, LinkedIn profiles from individual researchers, Catholic University of Croatia social media platforms, and the individual, personal profiles of the researchers (see the Participants section). The questionnaire was distributed for a period of a month and a half, from mid-November to the end of December 2021, stopping around the 1-year mark since the earthquake. Given that a previous study found that the PTE effect is stable for at least 1 year (*Li et al., 2010*), we judged this timeframe to be appropriate for the purposes of this study.

Following the initial stage of data collection, we were left with a small number of participants in the yellow ($n = 19$) and red ($n = 7$) categories. As there was a sizable difference in the number of participants between these two categories and the white and green category, we decided to resume the data collection for a period of 2 weeks in June of 2022. The data collection lasted until mid-June, and it was distributed specifically with the aim of collecting more participants belonging to the red and yellow categories. The questionnaire was distributed through contacting doctors, nurses, and pharmacists in the Petrinja area where the earthquake occurred, as well as through organizations aiding the earthquake victims. This additional data collected resulted in eight additional respondents, of which two were in the yellow category, three in the green category, and three in the red category.

## Statistical analysis

The statistical analysis was performed on a personal computer using the Statistical Package for Social Sciences (SPSS) software package. To test the first hypothesis that we could replicate the PTE effect, we performed a one-way ANOVA with level of devastation as an independent factor with four levels (*i.e.*, four colored stickers; white, green, yellow, and red). The five questions relating to the degree of post-earthquake concern (*Li et al., 2009*) were the dependent variables. The scores on these five questions were analyzed independently, as originally done by *Li et al. (2009)*. We corrected for multiple

comparisons for the five items of the original scale by *Li et al. (2009)*, using the Bonferroni procedure (corrected alpha level = 0.01). For each ANOVA, we checked the homogeneity of variance using Levene's test. In case of a violation of the assumption of homogeneity of variance, Welch's ANOVAs were used.

To test the second hypothesis entailing that we could replicate the PTE effect using a validated questionnaire, we performed the same analysis with the overall scores of the three DASS-21 (*Ivaković, 2019*; *Lovibond & Lovibond, 1995*) subscales as the dependent variables. Similarly, we corrected for multiple comparisons for the three subscales of the DASS-21 scale using the Bonferroni procedure (corrected alpha level = 0.017).

For the third hypothesis entailing that coping self-efficacy would mediate the relationship between the level of devastation and psychological distress, we performed mediation analyses using the PROCESS (*Hayes, 2022*) macro in SPSS. In the models, level of devastation was the independent variable, responses on the DASS-21 the dependent variables, and the coping self-efficacy and earthquake-specific coping self-efficacy scales as the mediating variables, included separately. Mediation was evaluated by inspecting the coefficient of the indirect pathway. If the 95% confidence interval did not include zero, this was considered as evidence for a significant mediation effect.

## RESULTS

### Handling of missing data and descriptive statistics

When measuring the post-earthquake health and safety concerns, we noted a different number of missing values per each question. For the illness probability question, there were no missing values, but for the other four questions there was a maximum of 10 missing responses. The data were analyzed for each question on the available responses (*i.e.*, excluding missing values). No missing data was recorded for the DASS-21 scale (*Ivaković, 2019*; *Lovibond & Lovibond, 1995*). Tables 1 and 2 show the mean values and standard deviations of level of post-earthquake health and safety concern measures and the DASS-21 (*Ivaković, 2019*; *Lovibond & Lovibond, 1995*) scores according to the level of devastation group.

### Post-earthquake health and safety concerns (replication *Li et al., 2009*)

For the first question concerning illness probability, the Levene's test was significant, $F(3, 312) = 1.179$, $p = 0.043$. Therefore, we used the Welch ANOVA. The Welch test showed that there is no significant difference between groups (*i.e.*, red, yellow, green, or white sticker) in terms of assessing the levels of probability of an illness spreading in the area hit by an earthquake, $F_{Welch}(3, 23.033) = 1.099$, $p = 0.364$.

Second, regarding the question about safety measures, Levene's test was also significant, $F(3, 305) = 2.903$, $p = 0.035$. The Welch ANOVA showed that there is no significant difference between groups in their approximation of safety measures residents of areas hit by an earthquake should take, $F_{Welch}(3, 31.566) = 0.446$, $p = 0.722$.

Third, for the question about the approximation of the number of medical doctors needed per 1,000 residents of the area hit by an earthquake, Levene's test showed that the assumption of homogeneity of variance was met, $F(3, 311) = 2.138$, $p = 0.095$. The one-way

**Table 1 Mean levels of post-earthquake health and safety concern measured with the original items from *Li et al. (2009)* according to damage levels groups.**

| | White sticker ($n$ = 151) | | Green sticker ($n$ = 134) | | Yellow sticker ($n$ = 21) | | Red sticker ($n$ = 10) | |
|---|---|---|---|---|---|---|---|---|
| | M | (SD) | M | (SD) | M | (SD) | M | (SD) |
| Illness probability (0–100%) | 46.08 | (29.14) | 41.67 | (27.12) | 45.52 | (37.49) | 31.50 | (28.97) |
| Safety measures (0–100) | 41.36 | (33.05) | 43.42 | (33.41) | 52.20 | (43.57) | 38.10 | (38.53) |
| No. doctors (per 1,000 residents) | 55.67 | (98.12) | 77.57 | (149.41) | 57.14 | (71.75) | 90.00 | (155.40) |
| No. psychologists (per 1,000 residents) | 93.05 | (150.30) | 144.26 | (237.19) | 237.00 | (342.53) | 182.80 | (312.03) |
| Medicine dose (0–100 mg) | 47.06 | (38.44) | 47.98 | (39.31) | 52.62 | (40.75) | 40.60 | (38.36) |

**Table 2 Mean levels of depression, anxiety, and stress measured with the DASS-21 according to damage levels groups.**

| | White sticker ($n$ = 151) | | Green sticker ($n$ = 134) | | Yellow sticker ($n$ = 21) | | Red sticker ($n$ = 10) | |
|---|---|---|---|---|---|---|---|---|
| | M | (SD) | M | (SD) | M | (SD) | M | (SD) |
| Depression | 4.53 | (4.05) | 5.96 | (4.86) | 7.62 | (5.10) | 10.40 | (6.98) |
| Anxiety | 3.62 | (4.12) | 4.88 | (4.31) | 7.43 | (5.25) | 8.60 | (7.23) |
| Stress | 6.42 | (4.59) | 8.04 | (4.65) | 10.19 | (6.07) | 11.80 | (7.02) |

ANOVA test showed that there is no significant difference between groups in their approximations of the number of medical doctors needed, $F(3, 311) = 0.908$, $p = 0.438$, $\eta p^2 = 0.009$.

Fourth, for the question about the approximate number of psychologists that is needed after the earthquake, Levene's test was significant, $F(3, 309) = 9.732$, $p < 0.001$. The Welch ANOVA shows that there is no significant difference between groups in their approximation of the number of psychologists needed, $F_{Welch}(3, 30.962) = 2.588$, $p = 0.071$.

Lastly, for the question about the approximate appropriate medicine dosage, Levene's test was not significant, $F(3, 302) = 0.290$, $p = 0.833$. The one-way ANOVA showed there is no significant difference between groups in their approximation of the dosage of the medicine needed, $F(3, 302) = 0.237$, $p = 0.870$, $\eta p^2 = 0.002$.

Taken together, the results show we failed in replicating the PTE effect based on the original questions used by *Li et al. (2009)* as none of the five questions showed a significant difference between the different level of devastation-groups (see Table 1).

## DASS-21 and Psychological Typhoon Eye effect

For the DASS-21 depression subscale, Levene's test was significant, $F(3, 312) = 4.620$, $p = 0.004$. The Welch ANOVA showed that there is a significant difference between the level of devastation groups in their depression levels, $F_{Welch}(3, 31.725) = 5.618$, $p = 0.003$. Games-Howell *post hoc* test revealed the group with white stickers (see Table 2) showed significantly lower levels of depressive symptoms than the group with green stickers

($p = 0.040$) (Table 2). There were no significant differences between the other groups ($p$-values > 0.05).

For the anxiety subscale of the DASS-21, Levene's test was significant, $F(3, 312) = 5.107$, $p = 0.002$. The Welch ANOVA showed that there is a significant difference between the level of devastation groups in terms of levels of anxiety reported by the participants, $F_{Welch}(3, 31.584) = 5.570$, $p = 0.003$. The Games-Howell *post hoc* test revealed that the white sticker group showed significantly lower levels of anxiety than the red sticker group ($p = 0.02$) (Table 2). No significant differences were found between the other groups ($p$-values > 0.05).

Finally, for the DASS-21 stress subscale, Levene's test was not significant, $F(3, 312) = 2.629$, $p = 0.05$. The one-way ANOVA showed that there is a statistically significant difference in stress levels according to the level of damage experienced in the earthquake $F(3, 312) = 7.932$, $p < 0.001$, $\eta p^2 = 0.071$. Bonferroni *post hoc* test revealed that the group with white stickers showed significantly lower levels of stress than the groups with green stickers ($p = 0.03$), yellow stickers ($p = 0.005$) and red stickers ($p = 0.004$) (Table 2). There were no statistically significant differences between other groups ($p$-values > 0.05).

Taken together, for each of the subscales of the DASS-21, we did not find the PTE effect. Instead, respondents least affected by the earthquake (*i.e.*, green, and white stickers) consistently showed the lowest level of psychological distress, and respondents most affected (*i.e.*, yellow and red stickers) showed the highest levels of psychological distress (see Table 2). This pattern of results does not correspond with the PTE effect, but rather indicates a ripple effect (*i.e.*, highest depression, anxiety, and stress in the most affected groups, and gradually lower depression, anxiety, and stress in least affected groups).

## Mediation analyses for the Psychological Typhoon Eye effect

The third goal of this study was to examine the working mechanisms of the PTE effect. The results of the ANOVA performed on the original five questions about post-earthquake levels of health and safety concerns and the DASS-21 scale showed that the PTE effect was not present in our sample. However, the ANOVA performed on the DASS-21 scale did show significant differences between the devastation groups suggesting a ripple effect. Therefore, we still decided to perform the mediation analysis to see if coping self-efficacy (*Ivanov & Penezić, 2002*; *Schwarzer, 1993*) and/or earthquake-specific coping self-efficacy (*Sumer et al., 2005*) mediated this ripple effect.

First set of analyses were done using general coping self-efficacy as the mediator and level of devastation and the independent variable. Depression, anxiety, and stress were used as the dependent variables. We focus here on reporting the indirect pathway in the mediation model, as this crucially tests the mediation hypothesis. For all the dependent variables, no significant mediation was observed: indirect effect for depression = 0.031 (95%CI [−0.142 to 0.212]), indirect effect for anxiety = 0.018 (95%CI [−0.092 to 0.125]), and indirect effect for stress = 0.030 (95%CI [−0.137 to 0.209]). This indicates that general coping self-efficacy did not mediate the effects of level of devastation on depression, anxiety, or stress.

In the second set of analyses, the earthquake-specific coping self-efficacy was used as a mediator. Again, level of devastation was the independent variable and depression, anxiety, and stress were the dependent variables. For brevity, only the indirect pathways are reported here, as this is the crucial test for establishing mediation. First, the indirect effect of earthquake-specific coping self-efficacy on depression was significant (indirect effect = 0.715, 95%CI [0.380–1.638]). Second, the indirect effect of earthquake-specific coping self-efficacy on anxiety was also significant (effect = 0.684, 95%CI [0.378–1.035]). Third and final, the indirect effect of the earthquake-specific coping self-efficacy on stress was also significant (effect = 0.826, 95%CI [0.457–1.224]). Taken together, the effect of level of devastation on depression, anxiety, and stress was mediated by earthquake-specific coping self-efficacy, but not general coping self-efficacy.

## DISCUSSION

The goals of this study were threefold: The first goal was to replicate the original PTE effect observed in the 2008 Wenchuan Earthquake (*Li et al., 2009*) in the context of the 2020 Petrinja earthquake using the five questions about post-earthquake levels of health and safety concerns used in this prior study by *Li et al. (2009)*. The second goal was to try and replicate the findings of the PTE effect using the validated DASS-21 scale (*Ivaković, 2019*; *Lovibond & Lovibond, 1995*). The third aim was to explore mediation of the PTE effect by general and earthquake-specific coping self-efficacy.

First, contrary to our expectations, we did not replicate the PTE effect, regardless of using the original five items used by *Li et al. (2009)* in their initial study or the DASS-21 to measure psychological distress. Instead, we found no differences between the level of devastation groups based on the items used by *Li et al. (2009)* and we observed a ripple effect pattern for the DASS-21 subscales. One possible reason for this failure to observe the PTE effect might be because it is not a robust phenomenon. Indeed, several prior studies mentioned mixed results when investigating this phenomenon (*e.g.*, *Huang et al., 2020*; *Lateef, Alaggia & Collin-Vézina, 2021*; *Gao, Chen & Zou, 2022*). Possibly, the effect could be sensitive to subtle differences in the context of the natural disaster (*e.g.*, extent of the devastation, which was much graver in the 2008 Wenchuan earthquake) or in the design of the study. Regarding the latter, *Li et al. (2009)* used geographic location for the operationalization of the epicenter and periphery of the earthquake, whereas we used the extent of suffered damage to people's houses as an approximation of closeness to the epicenter of the earthquake. Possibly this might explain the different observed pattern between our study and the study by *Li et al. (2009)*. Importantly, we consider our method of operationalization of level of devastation experienced by the earthquake to be a more ecological valid than the operationalization by *Li et al. (2009)*, as earthquakes are natural phenomena that are not restricted by administrative borders.

The failure to replicate the PTE effect in our sample may also be attributed to the cultural differences in the public response to the earthquake or other emergencies. In China, where the effect was first investigated, the collectivistic culture emphasizes the collective public response to a natural disaster or an emergency (*Zhang et al., 2020*). Thereby, people in the epicenter of an earthquake or public experience may experience

more social support, and consequently less psychological distress, in collectivistic countries. Croatia, on the other hand, is considered an individualistic country, where these collective responses following the earthquake might have not been so prominent or emphasized. This role of cultural values in determining responses to natural disaster is speculative but may warrant further research.

However, probably the most important reason for not observing the PTE effect in our study was the use of the five items used by *Li et al. (2009)* in their original study into the PTE effect. The use of these questions resulted in several missing values because of participants either skipping the questions or indicating they do not know or understand how to answer them. That is, respondents often struggled to indicate how many doctors should be allocated to a certain area or how many milligrams of medication should be prescribed. Questions that result in such (non-)responses are considered overly complex and confusing, which hampers their validity to appropriately measure psychological constructs (*Rosellini & Brown, 2021*). Furthermore, these items by *Li et al. (2009)* were created to examine health and safety concerns. However, psychological distress entails other aspects than concerns, such as physiological symptoms (*e.g.*, sleeplessness), emotional changes (*e.g.*, feeling sad), and behavioral signs (*e.g.*, reduced social activities). Hence, to capture the full width of the 'psychological' changes in the PTE effect, better validated assessment tools should be used. All in all, future studies aiming at replicating the study by *Li et al. (2009)* should utilize their items with caution, and ideally alongside other previously validated questionnaires.

Second, in contrast to not being able to observe the PTE effect, we found clear evidence for a ripple effect (*i.e.*, highest psychological distress in the most affected areas, and less distress in lesser affected areas) using the DASS-21 (*Ivaković, 2019*; *Lovibond & Lovibond, 1995*). This observation corresponds to many other findings in the literature, indicating that those most affected by a natural disaster or public emergency show the most psychological distress (*e.g.*, *Hoven et al., 2005*; *Rehdanz et al., 2015*; *Yáñez et al., 2020*). The reasons for why exactly some previous studies observed a PTE effect remain not entirely clear and warrant further research. As we indicated above, previous observations of the PTE effect could be attributed to the exact operationalization of geographical distance to the epicenter, cultural differences (*i.e.*, collectivistic *vs.* individualistic), or the use of non-validated items with high measurement error, producing spurious results. Collectively, we believe that skepticism is warranted for the PTE effect and the ripple effect should probably be seen as the standard response to natural disasters and public emergencies.

Third, for our hypothesis regarding general and earthquake-specific coping self-efficacy as mediators of the PTE effect, we could not address this hypothesis because the PTE effect was not present in our study. Instead, we decided to perform the mediation analysis aiming to disentangle the mediating effect of general and earthquake-specific coping self-efficacy for the ripple effect that we observed. We found that general coping self-efficacy did not mediate the relationship between levels of damage and depression, anxiety, and stress. However, earthquake-specific coping self-efficacy did show a mediating effect between the level of damage and depression, anxiety, and stress. This suggests that the higher feelings of

depression, anxiety and stress in higher devastated areas are closely related to the earthquake-specific experiences and coping levels. In other words, the distress of participants from the highest devastated areas in our study was related to their feelings of not being able to cope the effects of the earthquake. This can be informative for rescue workers and policy makers, whereby a focus on earthquake-specific coping (or specific to any other disaster or emergency), rather than focusing on general coping skills, can be helpful to overcome psychological distress by those most affected by the disaster (*e.g.*, reassuring people that their homes can be made structurally safe against earthquakes).

There are several limitations to this study that require consideration. The first limitation concerns the small sample of respondents in the yellow and red groups. Indeed, as mentioned before, out of 316 participants, only 10 indicated their home was severely damaged (red sticker) and only 21 indicated that there was substantial damage to their home (yellow sticker). Most of the participants were not in the most earthquake-devastated areas or their houses did not sustain any significant damages. The lack of information on those who sustained severe material damages might not give a proper representation of the psychological distress experienced by this group. Furthermore, the lack of data points in the most extreme categories resulted in a loss of statistical power. However, reaching those displaced by an earthquake was challenging and finding even this amount of people offers a unique perspective on those who suffered significant losses due to the earthquake. Furthermore, the fact that an opposite (*i.e.*, ripple) effect was observed for the DASS-21 suggests that sufficient statistical power was achieved to detect this alternative effect. A second limitation relates to the non-representativeness of the sample. Most of our participants were highly educated (*i.e.*, obtained a bachelor's degree or higher), likely due to recruitment in part taking place through university connections. This limits the extent to which the results can be generalized to the general population, though it is not necessarily expected that the PTE or ripple effect depend on level of education. A third limitation concerns the original five questions utilized by *Li et al. (2009)*. While the translation of these questions was performed in a thorough and detailed manner, the nature of the questions might have not been properly understood. Furthermore, one of the questions asked for participants' estimation on the probability of an epidemic disease spreading after the earthquake, which could have caused confusion when answering because at that time Croatia was in lockdown for the COVID-19 pandemic. Because of this context, the pandemic could have interfered with at least one of the questions. A fourth limitation is that some relevant constructs were not included in our study to not overburden participants, such as experience of other past traumatic events, pre-earthquake mental health problems, or economic impact of the earthquake. It would be interesting to investigate in future work whether these constructs may partially account for the PTE or ripple effect. Finally, a fifth limitation is the 1-year time lag from the time when the earthquake occurred and when the survey was distributed. This could have played a role in the results we obtained. While *Li et al. (2010)* did show that the PTE effect was robust for a whole year, constant aftershocks and reminders of unrepaired material

damages could have intensified the symptoms of depression, anxiety, and stress in the population that belonged to the red group. Because of the time elapsed it is impossible to know the initial responses and levels of health and safety concerns experienced by participants at the initial time of the Petrinja earthquake.

Despite the limitations, this study still provides unique data and a valuable insight into post-earthquake levels of health and safety concerns, as well as levels of depression, anxiety, and stress experienced by those in the epicenter and periphery of the 2020 earthquake in Petrinja, Croatia. It is, to our knowledge, the first study that aimed to replicate the PTE effect in the context of an earthquake following the original study by Li et al. (2009). It is also one of the rare studies that investigated the PTE phenomenon outside of the collectivistic countries in Asia, thereby offering a valuable extension of this work to different cultural contexts. In light of the ongoing replication crisis in science, attempting to replicate this relatively new phenomenon provides valuable information on its presence in different cultural settings and countries (Earp & Trafimow, 2015). Furthermore, our study is also one of the few studies examining psychological distress following the Petrinja earthquake, thereby contributing to the scarce research into the psychological impact of this traumatizing event in Croatia. Finally, we were able to utilize an ecologically valid system of operationalizing levels of devastation using colored stickers to label the epicenter of the earthquake, thus validly distinguishing between different proximities to the epicenter.

## CONCLUSIONS

In conclusion, this study was aimed at replicating the Psychological Typhoon Eye effect in the wake of the December 2020 earthquake in Petrinja, Croatia. We did not replicate the PTE effect using the original five questions about levels of post-earthquake health and safety concerns used by Li et al. (2009) in their original study, nor using the validated DASS-21 scale. Instead, the results showed a ripple effect using the DASS-21. Furthermore, we showed that earthquake-specific coping self-efficacy mediates the relationship between the level of damage and levels of depression, anxiety, and stress. This study is, to our knowledge, the first one that has attempted to replicate the PTE effect in the wake of an earthquake and has important implications for understanding the public response to natural disasters and allocation of psychological help to the survivors. We recommend that future studies examining the PTE effect should include at least one validated scale to measure psychological distress. Furthermore, future studies should explore the impact of different ways of operationalizing distance to a public emergency on the pattern of results (i.e., whether this relates to the observation of the PTE effect or a ripple effect).

## ACKNOWLEDGEMENTS

The authors would like to thank all the participants for investing their time into completing our questionnaires and the staff members of the Catholic University of Croatia for helping to recruit participants.

### Funding

The authors received no funding for this work.

### Competing Interests

The authors declare that they have no competing interests.

### Author Contributions

- Gaëtan Mertens conceived and designed the experiments, analyzed the data, prepared figures and/or tables, authored or reviewed drafts of the article, and approved the final draft.
- Marta Dürrigl conceived and designed the experiments, performed the experiments, analyzed the data, prepared figures and/or tables, authored or reviewed drafts of the article, and approved the final draft.

### Human Ethics

The following information was supplied relating to ethical approvals (*i.e.*, approving body and any reference numbers):

Ethical Review Board of the Tilburg School of Social and Behavioral Sciences

### Data Availability

The anonymized working file, SPSS output files, and an overview file of the used questionnaires in Qualtrics are available at DataverseNL: Mertens, Gaëtan; Dürrigl, Marta, 2024, "Psychological distress in the aftermath of the 2020 Petrinja earthquake, Croatia", https://doi.org/10.34894/UYKJBV, DataverseNL, V1.

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
