# Peer review of "A conceptual replication of the Psychological Typhoon Eye effect in the aftermath of the Petrinja earthquake in Croatia"

_PeerJ, doi:10.7717/peerj.18682_

## Round 0.1 · original submission · Major Revisions

Two reviewers have provided detailed comments. Please address them in an appropriate revision

Reviewer 1 ·

Basic reporting

Overall, this was a well-written paper.

The Abstract, under Methods, needs to explain how the PTE was examined in the present study (readers might be expecting comparisons of people living in the epicenter with people living away from the epicenter). Including citations in the Abstract is unorthodox, unless there is good reason. Also, please read the relevant comments below, and reconsider the combination of natural disasters with public emergencies (line 16) and the connection of the PTE effect after earthquake with psychological distress (line 17).

Experimental design

The Introduction was informative, however, a more critical take on the PTE was needed as well as more clarity about its connection with post-traumatic emotional distress. Li et al.'s suggestions followed the assessment of survivors’ generic concerns about health and safety after earthquake (which included asking lay people questions about how many doctors should be needed for every 1000 residents in the earthquake areas, or what dose of a fantastic medication would be needed to heal psychological wounds). Is there any empirical evidence for the connection between such beliefs and psychological distress after natural disaster? If there is, this needs to be evaluated and included in the Introduction. If not, this needs to be clearly stated and then relevant sections in the Abstract and the Introduction need to be revised. Also, the context of a public emergency (SARS, COVID, risk of lead-zinc mining) is very different from that in a post-earthquake setting, and the two stressors (public emergency and earthquake) are very different stressors. This has to be considered. For this reason, the Abstract (lines 16-17) and the Introduction (lines 76-78), which lump everything together, are a little confusing and could be misleading. The same applies to the sentence in lines 97-99 (‘…those most directly exposed to an emergency or natural disaster may feel that they acquired the necessary skills to mitigate these threats in the future’). If Li et al (or others) implied that the PTE effect refers to post-earthquake psychological distress being less severe in the most affected areas and more severe in less affected areas, then this needs to be clearly stated, or the phrase 'psychological distress' (line 78) needs to be reconsidered. For clarity, in the Abstract and the Introduction, the authors may wish to separate the results of studies that reported the PTE effect explicitly in relation to natural disasters and particularly in terms of whether these studies implied less emotional distress (rather than generic health and safety beliefs) closer to the epicenter compared to further away. A more thorough critique of the ‘psychological’ parameter in PTE would make the paper stronger.

The study used convenience sampling and the recruitment of an adequate number of participants was well guided by a power calculation. Approximately 90% of the participants reported no damage or slight damage to their houses. This point was considered carefully in the Discussion. Recruitment seems to have taken place primarily in the University (lines 152-153), involving University staff and students, and this may explain why approximately 63% of the participants were highly educated (at bachelor's or master level). This point should also be considered briefly when discussing the limitations of the study.

The measures used were appropriate and well discussed. However, the study included few measures and thus other important variables were not considered (e.g., experience of other past traumatic events, pre-earthquake mental health problems, economic impact of the earthquake, etc.), which is another limitation of the study that needs to be mentioned in the relevant section. Also, the Earthquake-specific Coping Self-efficacy items systematically assessed the survivors beliefs about ‘this earthquake’, not future earthquakes. Responses to the Earthquake Risk Perception scale do not necessarily tap the survivors’ self-efficacy for future earthquakes. It would be good to see some discussion about whether this may have led to loss of important information.

Validity of the findings

Most of the study's limitations were fairly addressed in the Discussion. Additional limitations and elements which need to be also considered have been mentioned above.

The results were discussed in a clear and well-organized way.

Additional comments

The word 'materialistic' in relation to damage, needs to be replaced throughout the manuscript with the word 'material' (i.e., material damage).

Reviewer 2 ·

Basic reporting

First of all, I would like to thank the author(s) for their efforts. The study seems to focus on an important issue, but I think there is a problem with the structuring of the study. Therefore, my suggestions regarding the study are presented below.

In the abstract section of a study, it is generally recommended not to provide citations. The abstract should primarily begin with the purpose of the study and then present other data.

Although the purpose of the study seems to be revealed at the end of the introduction section (Lines 120-137), there is a complex situation. There is no need to include the scales used in this section. The contribution to the gap in the literature can be presented in bullet points. Your hypotheses should be stated clearly. Additionally, more literature support is needed to highlight the study's unique value.

It was evaluated that the information in the method section of the study was repetitive and confusing. This section should first describe the study design. Then, the inclusion and exclusion criteria for participants, data collection tools, data collection procedures, analysis, and ethical considerations can be included. The sample size was determined to be 280 based on power analysis. Was a reference article considered for this?

Seven participants were excluded because they were not of legal age. Is there no information about consent in the information section of the survey form?

In the results section, there is no need to reiterate why certain analyses were conducted. The findings related to the hypotheses should be presented clearly.

In the discussion section, repetitive scale information can be removed. This section should briefly address the aim and then compare the findings with the literature.

In the conclusion section, results obtained from the study's findings should be provided, and the author(s) should offer recommendations. However, this is not reflected in the current version of the study. Therefore, the conclusion section should be revised.

Experimental design

Suggestions are provided in the above comments section.

Validity of the findings

Suggestions are provided in the above comments section.

---

## Round 0.2 · Minor Revisions

Please improve your write up as per comments soonest. Thanking you

Reviewer 1 ·

Basic reporting

The Abstract is still not clear about the design of the study and the ways through which the PTE was replicated in the study.
The introduction is somewhat improved in terms of criticality (with the addition of the lines 98-109). However, even though the paper is well-written, most of my previous comments still hold. There is still lack of sufficient clarity in the introduction and this is important, because it makes the justification for the present study appear weak. Nowhere in the Introduction does is say that Li et al (or others) implied that the PTE effect has to do with post-earthquake psychological distress (being less severe in the most affected areas and more severe in less affected areas). If they did, then this needs to be clearly stated. If they didn't, then this needs to be more explicit and in that case the present study does not really 'replicate' the Li et al suggestions, but explores something else. Thus the second goal of the present study ("...to try and replicate the findings of the PTE effect using the validated DASS-21 scale...") remains somewhat unjustified.

Experimental design

No further comments.

Validity of the findings

Previous comments raised questions about the conclusions relating to the final hypothesis of the study (..."we expected that coping self-efficacy will mediate the relationship between the level of experienced devastation and psychological distress."). Lines 504-506 (esp. the phrase "their feelings of not being able to cope with the potential threat of earthquakes") implies that what was assessed relates to estimated coping in future earthquakes. The manuscript needed more clarity there.

Additional comments

No further comments.

Reviewer 2 ·

Basic reporting

I salute the author(s) for their efforts. It has been observed that the suggestions presented within the scope of the study have been meticulously evaluated.

Experimental design

It was observed that the research design was appropriate and the suggestions presented were made.

Validity of the findings

The findings presented within the scope of the study were given using appropriate analysis methods.

Additional comments

I would like to thank the author(s) for their efforts in this study.

---

## Round 0.3 · accepted · Accept

Thank you for your revised manuscript which has been reviewed by myself and I am satisfied with the changes made according to comments with relevant refeences.